# Sustainably powering wearable electronics solely by biomechanical energy

Jie Wang[1,2,*], Shengming Li[1,*], Fang Yi[1,*], Yunlong Zi[1], Jun Lin[2], Xiaofeng Wang[1], Youlong Xu[2] & Zhong Lin Wang[1,3]

Harvesting biomechanical energy is an important route for providing electricity to sustainably drive wearable electronics, which currently still use batteries and therefore need to be charged or replaced/disposed frequently. Here we report an approach that can continuously power wearable electronics only by human motion, realized through a triboelectric nanogenerator (TENG) with optimized materials and structural design. Fabricated by elastomeric materials and a helix inner electrode sticking on a tube with the dielectric layer and outer electrode, the TENG has desirable features including flexibility, stretchability, isotropy, weavability, water-resistance and a high surface charge density of $250\,\mu C\,m^{-2}$. With only the energy extracted from walking or jogging by the TENG that is built in outsoles, wearable electronics such as an electronic watch and fitness tracker can be immediately and continuously powered.

[1] School of Materials Science and Engineering, Georgia Institute of Technology, Atlanta, Georgia 30332, USA. [2] Electronic Materials Research Laboratory, Key laboratory of the Ministry of Education & International Center of Dielectric Research, Xi'an Jiaotong University, Xi'an 710049, China. [3] Beijing Institute of Nanoenergy and Nanosystems, Chinese Academy of Sciences; National Center for Nanoscience and Technology (NCNST), Beijing 100083, China. * These authors contributed equally to this work. Correspondence and requests for materials should be addressed to Z.L.W. (email: zhong.wang@mse.gatech.edu).

Wearable electronics, including smart fabrics, wearable light-emitting diodes, health monitoring and motion tracking, are rapidly rising fields in today's technologies[1–6]. From the perspectives of both the practice and aesthetic wearable devices, as well as their power units, are required to be small, lightweight, flexible and washable. There has been substantial progresses in reducing the power requirements of devices and increasing the energy densities of batteries, but these systems are still using rigid lithium ion batteries (LIBs) without a self-charging technique; therefore, the batteries are frequently need to be charged or replaced/disposed[7]. One of the most promising ways to address such issues is the employment of energy-harvesting technologies from the ambient environment for sustainable operation[8]. There are some possible energy-harvesting power sources, such as solar cells that harvest energy from sunlight and thermoelectric generators that produce electricity from a temperature gradient. However, they cannot ensure a continuous power supply for wearable devices due to the intermittency of sunlight and the low output of thermoelectricity from body heat. Therefore, an energy harvester that works continuously and permits high levels of electrical energy generation is required. To harvest the ubiquitous and constantly available mechanical energy, triboelectric nanogenerators (TENGs) were invented to generate electricity from ambient mechanical motion, such as rotary motion, vibration, oscillating motion and expanding/contracting motion[9–16]. Owing to its advantages of light weight, small size, high efficiency and a wide choice of materials, TENGs have been utilized for self-charging power systems[17], active sensors[18–20] and sustainable energy sources[21]. Several studies have been working to harvest energy from body motions in more convenient and efficient ways[22–25]. However, to harvest biomechanical energy as the sustainable source for wearable electronics remains a task to be solved, which would enhance the portability of the wearable electronics, because previous TENGs either failed to come up with sufficient electric output to sustainably power electronic devices or needed to be triggered by other machines[17].

In this work, we aim at sustainably powering wearable electronic devices with high-output TENGs and corresponding energy storage units purely by harvesting biomechanical energy from daily motion. To achieve the goal, three principles need to be considered: (1) The TENG in this work should have high output performance; (2) It should be easy for the TENG to harvest biomechanical energy from daily human motions; (3) The self-powered system should be wearable. Therefore, this wearable power source is realized via a TENG with a rationally designed helix-belt contact structure, which is flexible, stretchable, weavable, light weight, low cost and water proofing. The TENG reveals electrical outputs with a charge density of $250 \, \mu C \, m^{-2}$. The symmetric structure of the tube-like TENG guarantees stable performance when it harvests energy from multiple types of mechanical motion as triggered from various directions. The geometry of the TENG can be tailored to meet customers' requests due to its scalable fabrication process. Mounted under shoes or weaved into cloth, the TENG-tubes convert human motion such as walking or jogging into electricity, which immediately and sustainably powers wearable electronics such as an electronic watch and a fitness tracker with the combination of a supercapacitor or battery to form a self-charging power system.

## Results

**Structure and material design of the TENG.** As illustrated in Fig. 1a, the tube-like TENGs are assembled into cloth or shoes to drive wearable electronic devices. The TENG-tubes weaved into

textile have a diameter of 2–3 mm (Fig. 1b) while the TENG tubes mounted under shoes have a diameter of 6–7 mm (Fig. 1c). Wearable electronics, such as electronic watch and fitness tracker, can be immediately and sustainably powered by the energy generated by the TENG tubes during walking or jogging, without extra power management unit. This achievement is enabled by the high output performance of the TENG tube, which has optimized structure and material designs.

For a TENG, its short-circuit current as well as open-circuit voltage are both proportional to its triboelectric surface charge density; and its output power density is proportional to the square of its triboelectric surface charge density[26,27]. When the most negative material in the triboelectric series (fluorinated ethylene propylene) is utilized as the dielectric material, the measured charge densities are about 20, 40, 80, 134 and $219 \, \mu C \, m^{-2}$ with solid gallium, copper, horn-like polypyrrole (hPPy), liquid galinstan and liquid gallium serving as the counter partner material, respectively[28–30]. It can be seen that liquid metal leads to the highest output, followed by the flexible nanomaterial, then the solid metal. This phenomenon indicates that besides the distinction between the electron affinities of two triboelectric materials, the effectiveness of contact between two triboelectric surfaces also plays an important role in charge density enhancement. It has been found that an effective charge transfer process for contact electrification occurs when the contact distance between two surfaces is around intermolecular distance[31]. Therefore, the soft and flexible contact surfaces help improve the area for charge transfer in comparison with coarse solid surface, which is the foundation for the high output of the as-fabricated TENG in material aspect.

The detailed structure of the as-fabricated TENG is shown in Fig. 1d, e. It can be seen that the TENG is basically composed of two parts. One part is the dielectric layer and its back electrode (outer electrode), shaping into a tube. The other part is the belt-like inner electrode, which attaches to the interior surface of the tube and forms a helix extending along the inside of the tube. The two triboelectric parts are wrapped in another dielectric layer that protects them from contamination. The dielectric layer is made from silicone rubber, which has a strong tendency to gain electrons and possesses excellent flexibility and stretchability in all dimensions. The outer and inner electrodes are prepared from the mixture of silicone rubber, carbon black and carbon nanotubes (CNTs; Fig. 1f), which electric conductivity is $4.3 \, S \, m^{-1}$ and stretchability limit is 620% (Supplementary Fig. 1). The carbon black provides the basic conductivity; while the CNTs not only further enhances the conductivity due to its good conductivity under high strain but also enlarges the contact area by forming nanostructured surface[32]. Such rubber-based soft material endows the triboelectric layers with high contact intimacy.

**Working mechanism and electrical outputs of the TENG.** The working mechanism of the TENG-tube is briefly presented in Fig. 2a, which is based on a coupling of triboelectric effect and electrostatic induction[33]. When the tube is compressed, the inner electrode comes to contact with the dielectric layer. Since the dielectric layer has a higher ability to attract electrons, electrons on the inner electrode surface will transfer to the surface of the dielectric layer, resulting in a negatively charged dielectric layer surface and a positively charged inner electrode surface (i). Note that the triboelectric charges on the dielectric layer surface will be retained for a long period of time due to the nature of electrets. Once released, positive charges are induced on the outer electrode (ii); then electrons will flow from the outer electrode to the inner electrode through the load and finally reach an equilibrium (iii). As compressed again, electrons will flow back from the inner

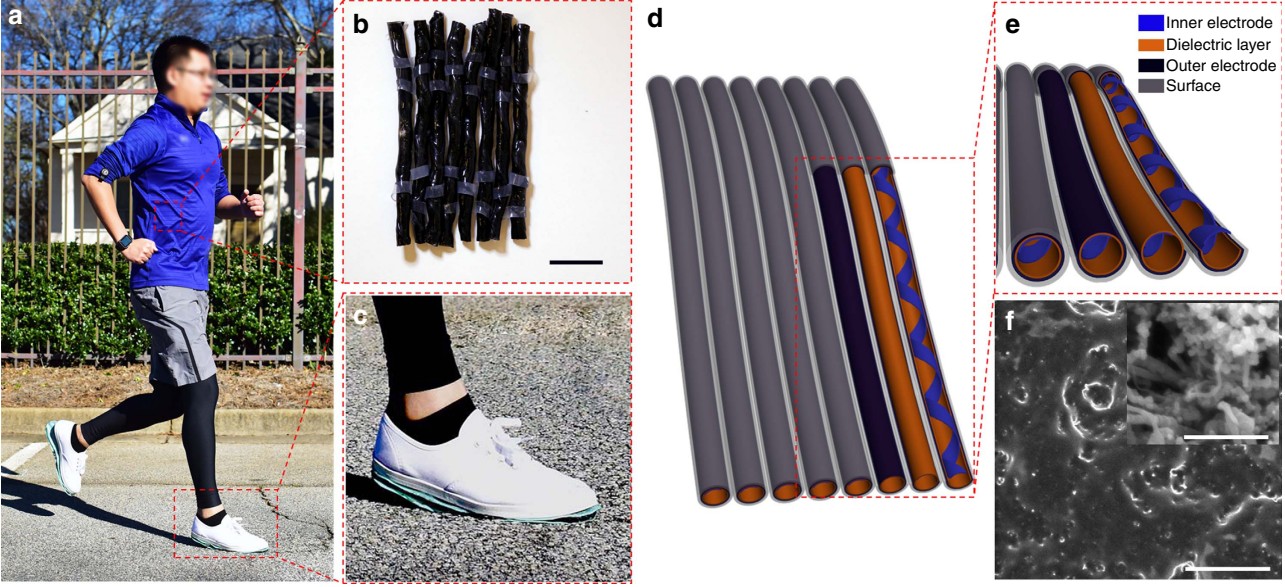

**Figure 1 | Overview of the TENG.** (**a**) TENG weaved into a coat and assembled under shoes to drive wearable electronics such as an electronic watch and a fitness tracker. (**b**) Photograph showing TENG tubes in diameter of 2–3 mm weaved into textile. (**c**) Photograph showing TENG-tubes fixed under a shoe. (**d**) Structure sketch of the TENG tubes. (**e**) Enlarged view of the TENG structure. (**f**) SEM image of the triboelectric electrode surface. The inset shows the SEM image of the carbon black/CNTs mixture, which is the conductive ingredient of the inner and outer electrodes. Scale bars in **b**, **f** and the inset in **f** are 1 cm, 5 μm and 500 nm, respectively.

electrode to the outer electrode (iv) and ultimately reach a new balance, until the inner electrode contacts the dielectric layer again (i). Thus, alternative current can be produced via periodically compressing and releasing of TENG tube.

The surface charge density ($\sigma_{SC}$) of the helix-belt structured TENG tube is boosted to be about 250 μC m$^{-2}$. This value is among the highest values reported[28–30] and is the basis of high output performance of TENG because it improves the energy harvested per operation cycle with higher short circuit current. Here the charge density of the helix-belt structured TENG is calculated by dividing the charges by the actual contact area of the inner electrode, which is only part of the total area of the inner electrode. This is because that there is a partial overlap of the helix inner electrode when the TENG is pressed flat and therefore the actual contact area is smaller. The detailed calculation for the contact area can be found in Supplementary Fig. 2 and Supplementary Note 1. Apart from the material aspect discussed above, it is found that the helix structure of the inner electrode has a considerably higher charge density than that of the traditional straight-layout structured TENG (Fig. 2d), which is only 110 μC m$^{-2}$ given the same size for contact areas. We have proceeded verifying experiments using the basic contact-separation TENG with the same materials to explore the reason for this difference. It is demonstrated that when the contact area enlarges, the transported charges slowly increase but the charge density drastically declines, with the charge density reaching to ~250 μC m$^{-2}$ when the size of the contacting surfaces is decreased to 5 × 5 mm$^2$ (Supplementary Fig. 3 and Supplementary Note 2). This indicates that the helix inner electrode is beneficial for high charge density by dividing one larger area as a whole into several smaller ones when compressed. This might be due to the improved contact effectiveness of the surfaces with the smaller size. And besides, when pressed from various directions, the helix-belt structured TENG presents a steady charge density (the red curve in Fig. 2e); while the straight-layout structured TENG shows an output that is direction sensitive, which even drops to zero when the oblique angle of the electrode to horizontal axis ($\theta$) is 90° (the blue curve in Fig. 2e).

The constant performance of the helix-belt structured TENG can be attributed to the fact that its symmetric structure results in an invariant contact area at different pressing directions.

To obtain an optimized charge density of the helix-belt structured TENG tube, three geometry parameters are introduced, namely the width of the inner electrode belt $d$, the oblique angle of the belt to horizontal axis $\theta$, and the width of the tube when it is flattened, $D$, that is, the half of its perimeter. Since the output surface charge ($Q_{SC}$) of a TENG usually increases with the contact area (Supplementary Fig. 3), the charge density is commonly used to compare the performance among TENGs. For the helix-belt structured TENG tube, once its total size is settled, it is ideal to achieve a maximal output charge coupled with the highest charge density through adjusting the geometry parameters. It is found that when $d$ remains at a constant value of 5 mm, the highest charge and charge density are simultaneously acquired at $\theta = 45°$ (Fig. 2f, g). When $\theta$ is set to be 45° (Fig. 2h, i), with the increase of $d$, the transferred charges increases first and then decreases, which is because part of the dielectric film is left untouched when $d$ is too small while part of the inner electrode is overlapping when $d$ is too large. The charges density decreases with the increasing $d$, and the optimal $d$ for charge and charge density is 5 mm for the TENG-tube with a determined size (Fig. 2j, k), where the obtained charge is 100 nC and charge density is 250 μC m$^{-2}$. Consequently, a relationship among these three parameters for an optimal performance of the helix-belt structured TENG can be derived, which is $d = D \times \sin \theta$.

**Extracting energy from diverse motion by the TENG.** Since the helix-belt structured TENG tube has excellent stretchability; besides the pressing motion, it can also harness energy from various other types of motion such as bending motion, twisting motion and lengthening motion. Figure 3 exhibits the test results of the helix-belt structured TENG (diameter: 7 mm) when harvesting energy in such kinds of motion. For the pressing motion (Fig. 3a–c), the charge density (~250 μC m$^{-2}$) and the peak value of the open-circuit voltage (~145 V) remain unchanged under different frequencies; while the peak value of

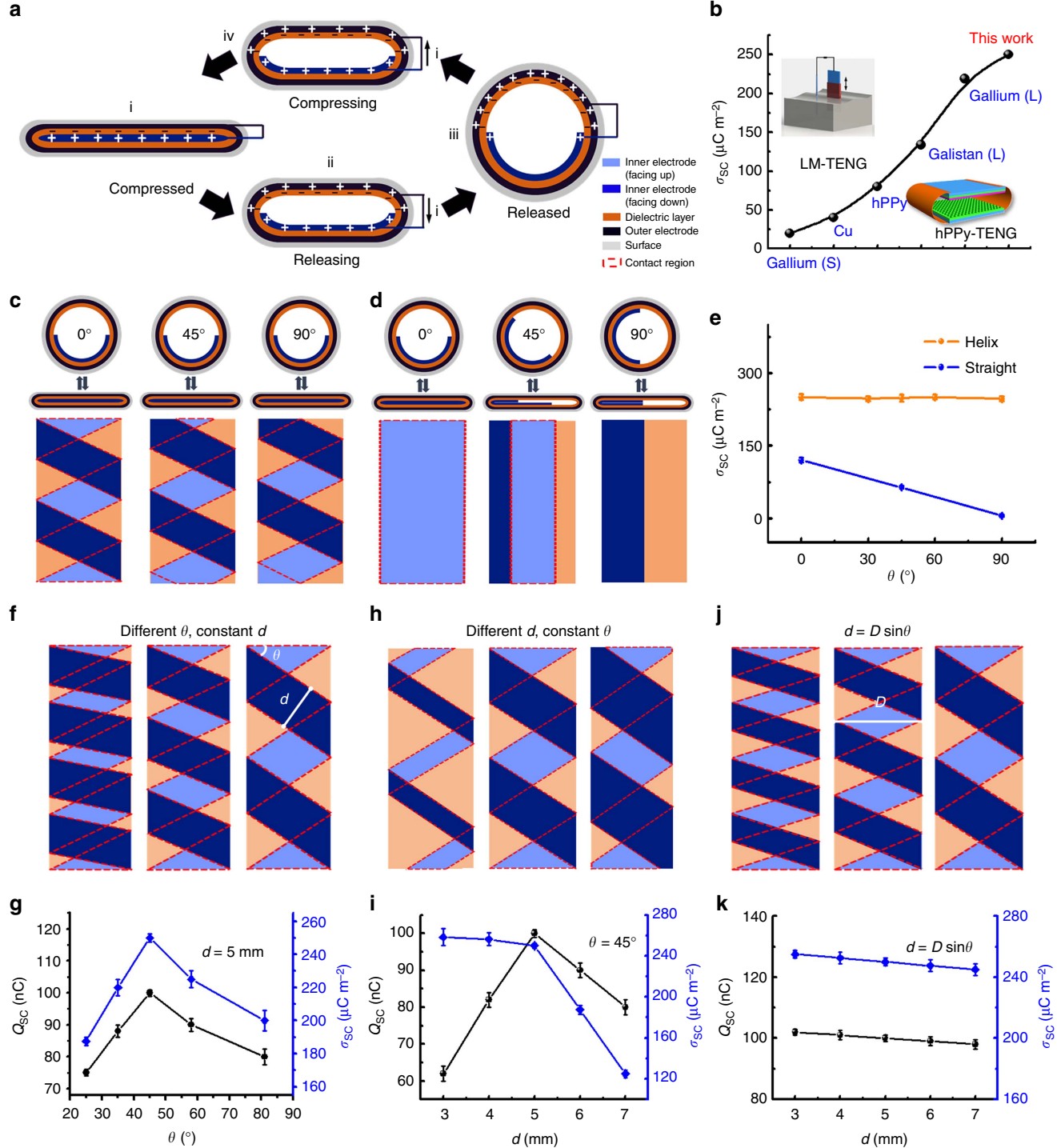

**Figure 2 | Working mechanism and output performance of the TENG.** (**a**) Working mechanism of the TENG. (**b**) Comparison of the output charges density ($\sigma_{sc}$) measured in this work with that in the previous works, insets in top-left corner and bottom-right corner show the structure of liquid-metal based triboelectric nanogenerator (LM-TENG) and horn-like polypyrrole based triboelectric nanogenerator (hPPy-TENG), respectively. (**c,d**) Schematic diagrams of the helix-belt structured TENG and the straight-layout structured TENG pressed at various directions. (**e**) $\sigma_{sc}$ of the helix-belt structured TENG and straight-layout structured TENG. (**f–k**) Schematic diagram and output charge ($Q_{SC}$) and $\sigma_{sc}$ of the TENG at various structure parameters, and the error bars are s.d.: (**f,g**) width of the triboelectric electrode ($d$) is 5 mm; (**h,i**) oblique angle of the electrode to horizontal axis ($\theta$) is 45°; (**j,k**) $d = D \times \sin\theta$, where $D$ is the width of the tube when it is flattened by pressing.

the short-circuit current density increases with the frequency, from 5 mA m$^{-2}$ at 2 Hz to 16 mA m$^{-2}$ at 10 Hz. Note that for the helix-belt structured TENG with the optimized geometry parameters as discussed above, when its total size further scales down, the output charges will vary but the output charge density will be maintained. For example, the output charges of a TENG tube with a diameter of 3 mm drops to 45 nC but the charge density still remains around 250 μC m$^{-2}$ (Supplementary Fig. 4).

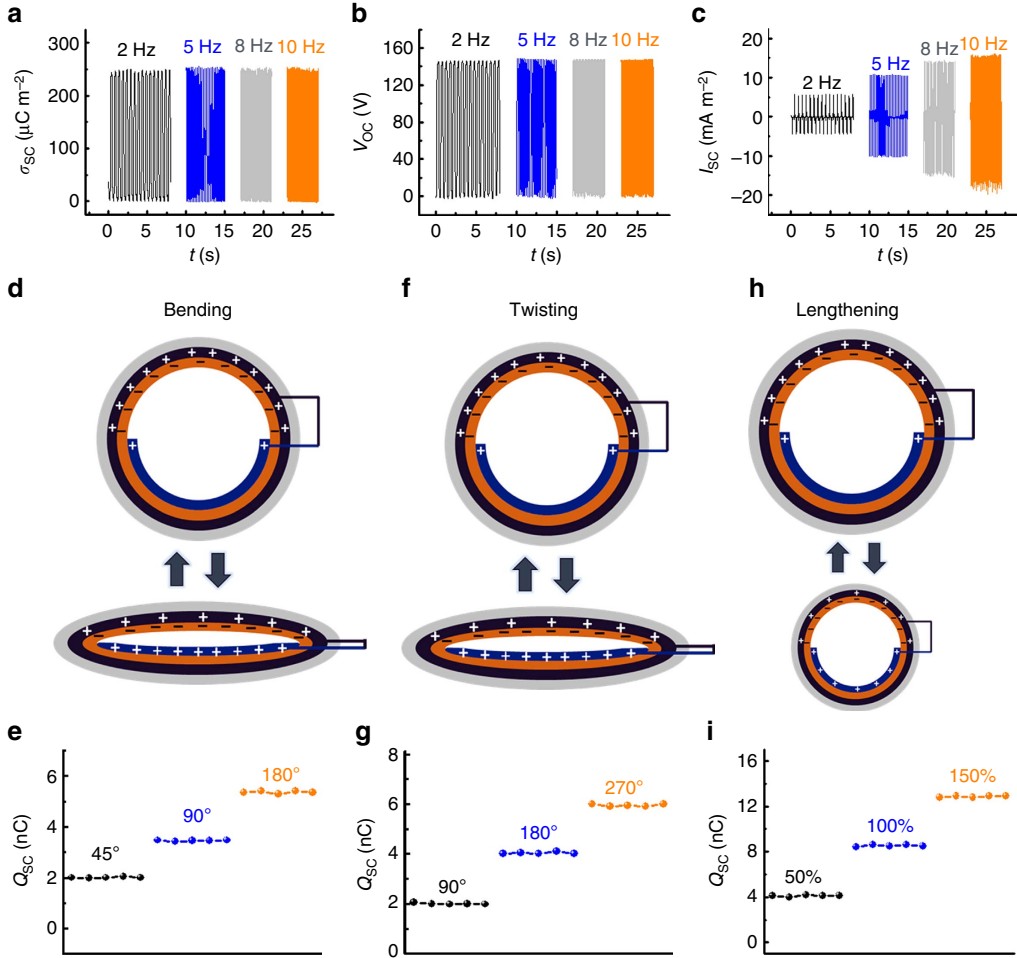

**Figure 3 | Electrical outputs of the TENG on diverse types of motions.** (**a**–**c**) (**a**) Charge density ($\sigma_{sc}$), (**b**) open-circuit voltage ($V_{OC}$) and (**c**) short-circuit current ($I_{SC}$) of the TENG at various pressing frequencies. (**d**–**i**) Working mechanism and output charge ($Q_{SC}$) of the TENG (**d**,**e**) under bending motion, (**f**,**g**) under twisting motion, (**h**,**i**) under lengthening motion.

The output charges can be linearly boosted by using multiple TENG tubes in parallel (Supplementary Fig. 5), for example, four TENG tubes generate a charge of 400 nC from the pressing motion. In addition, the TENG presents excellent performance stability, with its charge density, open-circuit voltage and short-circuit current maintaining steady after 3 million cycles of operation at a contact frequency of 10 Hz (Supplementary Fig. 6). These properties enable the TENG tubes to be an effective configurable power source such as the 'energy shoes' by assembling TENG tubes under/inside the shoes.

For the bending and twisting motion, the resulted deformation of the TENG tube is similar to the deformation caused by localized pressing. When bended from 45° to 180°, the charge of the TENG tube increases from 2 to 5.2 nC (Fig. 3d, e), while the open-circuit voltage increases from 10 to 25 V and the short-circuit current increases from 10 to 30 nA (Supplementary Fig. 7a,b). When twisted from 90° to 270°, the charge grows from 2 to 6 nC (Fig. 3f, g), while the open-circuit voltage increases from 10 to 30 V and the short-circuit current increases from 10 to 45 nA (Supplementary Fig. 7c,d). For the stretching motion, the tube length increases and the tube diameter decreases periodically. The charges increase from 4 to 12.5 nC as the tensile strain increases from 50 to 150%, while the open-circuit voltage increases from 20 to 63 V and the short-circuit current increases from 20 to 70 nA (Supplementary Fig. 7e,f).

**Applications of the TENG in wearable electronics**. The ability of accommodating to curvilinear surface and harvesting energy from diverse motion makes the helix-belt structured TENG-tube a desirable wearable power source. Moreover, since the TENG-tube is well packaged with a layer of silicone rubber, it is light-weight, waterproofing and anticorrosive, which can better suit the practical needs compared with traditional wearable power sources based on solid materials like metal. When immersed and swayed in water, the TENG tube maintains its high output charge density, and one single tube can sufficiently light up 32 light-emitting diodes (LEDs) by manually tapping it after taken out of water (Fig. 4a and Supplementary Video 1). When weaving 8 TENG-tubes in parallel into a shoe sole or a safety vest, LED warning signs on the vest such as 'CAUTION', 'PASS', and 'STOP' can be lighted up during walking or tapping the vest (Fig. 4b,c, Supplementary Video 2). The TENG tubes can be further integrated with a supercapacitor/LIB to form a self-charging power system (Fig. 4d). Note that when the power system is utilized to directly drive electronics, a certain number of TENG tubes can be employed according to the load of the electronics. Pressing by hand, it takes 70 s for 2 TENG tubes to charge the supercapacitor with the horn-like polypyrrole active material of 0.008 mg from 0 to 70 mV (Fig. 4e), where the equivalent galvanostatic current can be calculated as 1.4 µA (Supplementary Note 3). The morphology of horn-like polypyrrole and the

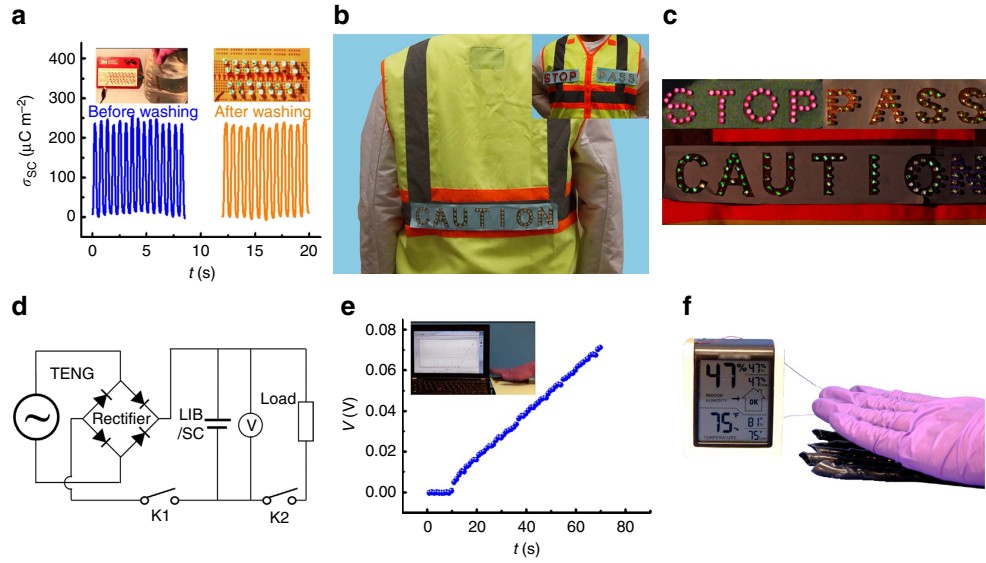

**Figure 4 | Applications of the TENG in emergency response and weather indication.** (**a**) Charge density of the TENG before and after washing in water. The left insert showing the TENG immerged into water, and the right insert showing that 32 LEDs were lighted by pressing the TENG after washed in water. (**b,c**) LED warning signs on a vest, i.e., 'CAUTION', 'PASS' and 'STOP' are lighted by pressing the TENG weaved into the vest. (**d**) Circuit diagram of the self-charging power system integrated by the TENG and SC/LIB. (**e**) Charging curve of SC by manually pressing two TENG-tubes. (**f**) A temperature-humidity meter is driven by manually tapping five parallel TENG tubes.

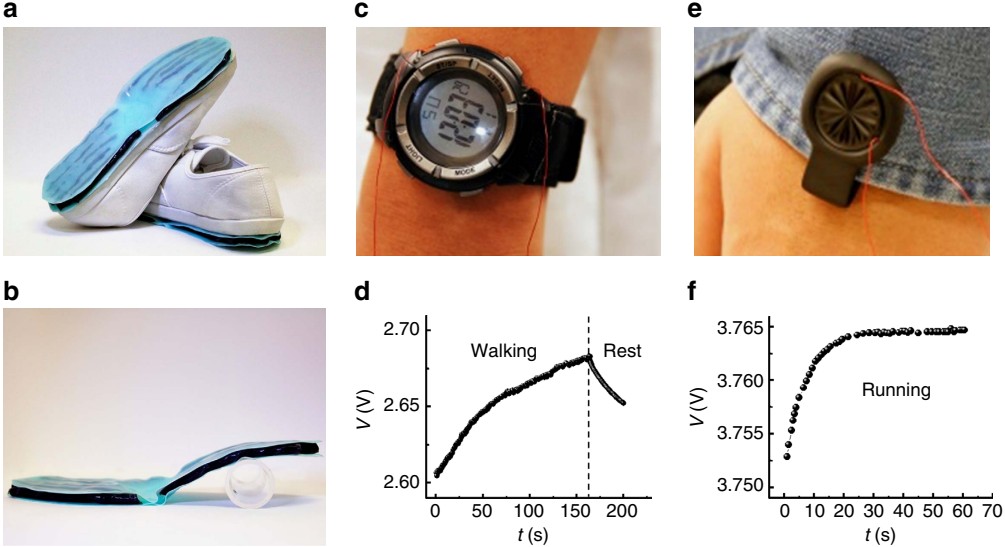

**Figure 5 | Sustainably powering wearable electronics during walking or jogging.** (**a**) Image of the 'energy-shoe'. (**b**) Image of the 'energy-outsole'. (**c**) An electronic watch is driven and (**d**) LIB in the self-charging power system is charged simultaneously by the 'energy-shoe' while walking. (**e**) A fitness tracker is driven and (**f**) LIB is charged simultaneously by the 'energy-shoe' when jogging.

charging–discharging curve of the supercapacitor are presented in Supplementary Figs 8 and 9. A temperature–humidity meter can be driven by manually tapping 5 TENG tubes connected in parallel (Fig. 4f, Supplementary Video 3).

With 40 tubes assembled under a pair of shoes (Fig. 5a), an electronic watch and fitness tracker called 'up move' can be immediately and sustainably powered sorely by walking and jogging (Fig. 5c–f). Note that the original batteries of the two electronic devices are replaced in advance by a home-made LIB with the $LiMn_2O_4$ active material of 0.5 mg that is specially designed for the TENG to ensure a fast charging rate. The morphology and X-ray diffraction pattern of $LiMn_2O_4$ is presented in Supplementary Fig. 10 and the charging–discharging

curve of the LIB is presented in Supplementary Fig. 11. While walking, the voltage of the battery keeps growing at the same time the electronic watch is powered, which indicates that the TENG tubes produces sufficient amount of energy to charge the LIB in addition to sustaining the consumption of the electronic watch (Fig. 5d, Supplementary Video 4). Once motion stopped, the voltage begins to decline as the LIB is discharged. The fitness tracker is driven by jogging, the self-charging power system is also able to support the electronic device and charge the LIB simultaneously (Fig. 5f, Supplementary Video 5). When the LIB is charged by pressing five TENG tubes connected in parallel at 10 Hz with a shaker, its voltage increases from 2.8 to 4.2 V during 7 h (Supplementary Fig. 11).

## Discussion

In summary, a sustainable operation of wearable electronics is achieved by a material-and-structure optimized TENG, which produces sufficient energy by converting human motion into electricity. From the structural aspect, the TENG adopts a rationally designed helix-belt structure that boosts the performance by dividing the electrode into several small areas for more sufficient contact, with a charge density of $250 \, \mu C \, m^{-2}$, which is among the highest reported to the best of our knowledge. Moreover, the helix-belt structured TENG is symmetric, which endows it with stable performance when triggered from various directions. From the material aspect, the rubber-based materials makes the TENG highly stretchable, light weight, anticorrosive and it can effectively extract energy from multiple deformations involving pressing, stretching, bending and twisting. The TENG is also waterproof and exhibits steady performance after being washed. Due to its potentially scalable fabrication process, the size of the TENG can be altered easily to fit various applications. The TENG may be applied for emergency response or weather indications by lighting up LED signs on a safety vest or driving a temperature–humidity meter with the energy harnessed from human motion. When integrated in a self-charging power system with a supercapacitor or LIB, a pair of outsoles assembled with the TENG can immediately and sustainably drive electronic devices such as an electronic watch and fitness tracker, where the energy harvested from only walking and jogging can simultaneously drive the wearable electronic device and charge the battery. This work may represent an advance in the development of wearable power sources and could offer new design options for self-powered systems.

## Methods

**Fabrication of the TENG.** The liquid silicone rubber was obtained by mixing the base and cure (1:1 by volume ratio) of the silicone rubber (Ecoflex 00–30) in a beaker, then it was added the mixture of conductive carbon black and CNTs (2:1, weight ratio). After they were mixed uniformly, carbon black/CNTs@silicone rubber, was smeared over a piece of acrylic plate preprocessed with a release agent and cured at a temperature of 30 °C for 5 h, a soft conductive electrode was obtained. The conductive electrode can be used as not only triboelectric electrode after it was cut to belts with a width of 5 mm, but also back electrode after it was coated by the silicone rubber as dielectric layer.

**Fabrication of the supercapacitor and LIB.** The supercapacitor is fabricated to be symmetrical structure, with two identical micro/nano-structured hPPy electrodes as positive and negative electrodes, and 1 M KCl aqueous solution as electrolyte. The synthesis process of the hPPy is discussed in Supplementary Note 4. The supercapacitor was packaged by two pieces of Kapton films in sandwich structure. In the LIB, $LiMn_2O_4$ is used as positive electrode active material and graphite as negative electrode material, with 1 M $LiPF_6$ in ethylene carbonate/diethyl carbonate/dimethyl carbonate (EC/DEC/DMC 1:1:1 by weight ratio) as electrolyte. For fabrication of $LiMn_2O_4$ electrodes, the prepared powders were mixed with carbon black and polyvinylidene fluoride (70:20:10 by weight ratio) in N-methylpyrrolidinon. The slurry thus obtained was coated onto Al foil and then dried at 120 °C overnight in vacuum condition. The graphite negative electrode with copper as current collector was obtain from commercial sources and was used as received. The battery is packaged to be 2016 type coin cells with porous polyethylene film as separator. The synthesis process of $LiMn_2O_4$ is depicted in Supplementary Note 5.

**Data availability.** The data supporting the findings of this study is available from the corresponding author on request.

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

## Acknowledgements

This work was supported by the Hightower Chair Foundation, the 'Thousands Talents' program for a pioneer researcher and his innovation team and National Natural Science Foundation of China. (21274115).

## Author contributions

J.W., S.L., F.Y. and Z.L.W. conceived the idea, analysed the data and wrote the paper. J.W. and F.Y. designed the materials of the triboelectric nanogenerators. J.W. and S.L. optimized the structure of the triboelectric nanogenerators. Y.L. derived the theories for structure optimization. J.L. and Y.X. fabricated the supercapacitor and lithium ion battery. X.W. helped with the experiments. All the authors discussed the results and commented on the manuscript.

## Additional information

**Competing financial interests:** The authors declare no competing financial interests.

