## [Peer review file · Nature Communications]

Reviewers' Comments:

Reviewer #1 (Remarks to the Author)

In these days wearable electronics is attracting high attention among many researchers, and a seamlessly integrated power supply module with appropriate energy harvesting and storage functions is an important goal to achieve. In this manuscript, Wang and co-authors described optimized materials and design strategy for triboelectric nanogenerators (TEGs) and integrated them with other wearable device units. Use of carbon black (CB) and carbon nanotube (CNT)-based conductive elastomer and a rationally designed helix-belt structure for TENG electrodes give rise to significant advances in TEG performances such as high charge density of $250 \mu\text{Cm}^{-2}$. Also, the authors demonstrated various wearable self-charging power supply systems consisted of the TENG-built in outsoles and integrated supercapacitors (SCs)/lithium ion batteries (LIBs). The system successfully converts mechanical energy gained from walking or jogging to electrical energy, enabling immediate and continuous supply of electrical power to the wearable devices. Therefore, the reviewer recommends publication of this manuscript in Nature Communications. There are some minor issues that can be addressed before the publication.

Comment #1: The authors focused on introducing their unique ideas regarding materials and structures of the TEGs. And relatively speaking, detailed description on SCs and LIBs was not included. Following comments would be helpful for readers.

(1) The authors synthesized electrode materials for the SC and LIB cathode. However, more detailed data such as microstructure images (SEM or TEM images) of the horn-like polypyrrole electrode and LiMn_2O_4 were not provided. Additional XRD data of the synthesized LiMn_2O_4 can be included to prove its successful synthesis.

(2) In the Method part, the authors should give more specific and detailed description on fabrication of the LIBs such as composition (active materials, binder, and conducting agent) and components (current collectors and separator).

(3) The authors showed charging of the SCs and LIBs by the TEGs in Figure 4e, and Figure 5d and f. The charging time of the SCs and LIBs is directly affected by loading mass of the electrode materials of the SCs and LIBs. Therefore, the authors should provide accurate loading mass of the electrode materials of the SCs (horn-like polypyrrole) and LIBs (LiMn_2O_4).

Comment #2: In this work, the very soft silicone rubber (Ecoflex) was used as TENG materials. Therefore, the reviewer is wondering whether the material and structure of the TENG can endure the repeated foot stomping while walking or jogging.

Comment #3: The resolution of the inset image of Figure 1a seems to be low. The authors may provide images with the better resolution.

Comment #4: The electrical conductivity of TENG electrodes is one of the most important factors to determine the performance of TENGs. The authors should provide electrical conductivity of the mixture of silicone rubber with CB and CNT.

Comment #5: The authors should mention the output voltage and current of TENGs in the experiments (Figure 3 and Figure S5) because they can significantly vary with the applied pressure.

Comment #6: In Figure 5, the authors showed charging of the LIBs by the "energy-shoe" while walking and jogging in the limited voltage range. However, the charging curve (Figure 5f) seems

to be saturated at ~ 3.765 V. Therefore, an additional full charging curve of the LIBs in the voltage range of 2.75 - 4.24 V as shown in Figure S7 would be helpful.

Comment #7: The wearable energy harvesting and storage modules are typically connected with LEDs and/or sensors. Hopefully their monolithic integration with wearable LEDs/displays (e.g., Nature Communications 6, 7149, 2015; Science Advances 2, e1501856, 2016) and/or biosensors (e.g., Nature 529, 509, 2016; Nature Nanotechnology AOP, DOI: 10.1038/nnano.2016.38, 2016) will be a future research and development goal. These systems are still using rigid LIBs without self-charging and thereby frequently need to be replaced, although self-charging would dramatically enhance the portability. There have been reports that wearable LIBs or fabric-based SCs are charged by integrated TEGs to prolong the time of use. Some comments on this research direction in the introduction would be helpful to give guidance for the next research step.

Reviewer #2 (Remarks to the Author)

The authors describe an interesting approach to fabricate wearable energy device through the triboelectric effects. They claimed that the tube-like design offers advantages including flexibility, stretchability, isotropy, weavability, and water-resistance. They claimed the highest surface charge density achieved. The biomechanics energy from walking or jogging appears to be sufficient to drive wearable electronics such as electronic watch and fitness tracker. This work is an extension of the group's previous research. This review recommend publication after addressing the following issues:

(1) what is the durability of the devices? In particular, how will the long-term ambient moisture influence the device performance.

(2) the research group has reported their turboelectric devices in a number of previous publications. I wonder the key conceptual and methodological advances compared to their previous results.

(3) what is the stretchability limit?

(4) some typological errors, e.g. " $\square\text{Cm}^{-2}$ " in the abstract;

Point-by-point responses to the reviewers' comments

We sincerely thank the reviewers for their careful and thorough review, which are indeed very helpful to make the paper more solid and smooth. We have revised our manuscript very carefully in the light of their suggestions and comments.

The following responses have been prepared to address all of the reviewers' comments in a point-by-point fashion. (**Comments in black, responses in blue**):

Reviewer #1 (Remarks to the Author):

In these days wearable electronics is attracting high attention among many researchers, and a seamlessly integrated power supply module with appropriate energy harvesting and storage functions is an important goal to achieve. In this manuscript, Wang and co-authors described optimized materials and design strategy for triboelectric nanogenerators (TENGs) and integrated them with other wearable device units. Use of carbon black (CB) and carbon nanotube (CNT)-based conductive elastomer and a rationally designed helix-belt structure for TENG electrodes give rise to significant advances in TEG performances such as high charge density of $250 \mu\text{Cm}^{-2}$. Also, the authors demonstrated various wearable self-charging power supply systems consisted of the TENG-built in outsoles and integrated supercapacitors (SCs)/lithium ion batteries (LIBs). The system successfully converts mechanical energy gained from walking or jogging to electrical energy, enabling immediate and continuous supply of electrical power to the wearable devices. Therefore, the reviewer recommends publication of this manuscript in Nature Communications. There are some minor issues that can be addressed before the publication.

Response: We appreciate the praise and the valuable comments of this reviewer.

Comment #1: The authors focused on introducing their unique ideas regarding materials and structures of the TEGs. And relatively speaking, detailed description on SCs and LIBs was not included. Following comments would be helpful for readers.

(1) The authors synthesized electrode materials for the SC and LIB cathode. However, more detailed data such as microstructure images (SEM or TEM images) of the horn-like polypyrrole electrode and LiMn_2O_4 were not provided. Additional XRD data of the synthesized LiMn_2O_4 can be included to prove its successful synthesis.

Response: Many thanks for your suggestion. The SEM and TEM images of the horn-like polypyrrole electrode have been provided in Supplementary Figure 8, and the SEM/TEM images and XRD of LiMn_2O_4 have been provided in Supplementary Figure 10. The diffraction peaks in XRD are identified as a single-phase of spinel LiMn_2O_4 .

Supplementary Figure 8. SEM (a) and TEM (b) images of horn-like polypyrrole.

Supplementary Figure 10. SEM (a) and (b), TEM (c) images and XRD (d) of LiMn₂O₄.

(2) In the Method part, the authors should give more specific and detailed description on fabrication of the LIBs such as composition (active materials, binder, and conducting agent) and components (current collectors and separator).

Response: Many thanks for your comments. The details on fabrication of LIBs have been provided in the revised Method part as following:

In the lithium ion battery, LiMn_2O_4 is used as positive electrode active material and graphite as negative electrode material, with 1M/L LiPF_6 in EC/DEC/DMC (1:1:1 by weight ratio) as electrolyte. For fabrication of LiMn_2O_4 electrodes, the prepared powders were mixed with carbon black and polyvinylidene fluoride (70:20:10 by weight ratio) in N-methylpyrrolidinon. The slurry thus obtained was coated onto Al foil and then dried at 120 °C overnight in vacuum condition. The graphite negative electrode with copper as current collector was obtain from commercial sources and was used as received. The battery is packaged to be 2016 type coin cells with porous polyethylene film as separator. The synthesis process of LiMn_2O_4 is depicted in Supplementary Note 5.

(3) The authors showed charging of the SCs and LIBs by the TEGs in Figure 4e, and Figure 5d and f. The charging time of the SCs and LIBs is directly affected by loading mass of the electrode materials of the SCs and LIBs. Therefore, the authors should provide accurate loading mass of the electrode materials of the SCs (horn-like polypyrrole) and LIBs (LiMn_2O_4).

Response: Many thanks for your suggestion. The accurate loading mass of SCs and LIBs are 0.008 mg and 0.5 mg, respectively, which has been provided in the revised manuscript.

Comment #2: In this work, the very soft silicone rubber (Ecoflex) was used as TENG materials. Therefore, the reviewer is wondering whether the material and structure of the TENG can endure the repeated foot stomping while walking or jogging.

Response: Thank you for your comment. As shown in Wikipedia (https://en.wikipedia.org/wiki/Silicone_rubber), Silicone rubber is generally non-reactive, stable, and resistant to extreme environments and temperatures from -55 °C to +300 °C while still maintaining its useful properties. Due to these properties and its ease of manufacturing and shaping, silicone rubber can be found in a wide variety of products, including: automotive applications; cooking, baking, and food storage products; apparel such as undergarments, sportswear, and footwear; electronics; medical devices and implants; and in home repair and hardware with products such as silicone sealants.

Although the very soft silicone rubber (Ecoflex) was used in the TENG, its output performance remains stable after 3 million cycles, as shown in Supplementary Figure 6. Generally, the average daily walk of an adult is around 4,000 steps, i.e. the TENG under the outsoles will be pressed around 2,000 cycles a day. Therefore, the TENG can remains stably for at least 4-year walking. It should be noted that although the TENG tubes are fabricated under outsoles, the layer to contact the ground can adopt other plastic materials

used by general shoes. As shown in Figure 1 (c) and Figure 5 (b), the TENG tubes do not contact the ground directly.

Supplementary Figure 6. Stability of the TENG, (a) charge density as function of cycle number, inset shows the charge density of the TENG before and after 3 million cycles at 10 Hz, (b) open-circuit voltage and (c) short-circuit current before and after 3 million cycles at 10 Hz.

Comment #3: The resolution of the inset image of Figure 1a seems to be low. The authors may provide images with the better resolution.

Response: Many thanks for your suggestion. The inset image of Figure 1a has been replaced by the image with the better resolution.

Comment #4: The electrical conductivity of TENG electrodes is one of the most important factors to determine the performance of TENGs. The authors should provide electrical conductivity of the mixture of silicone rubber with CB and CNT.

Response: Thank you for your comment. The conductivity of the mixture is 4.3 S m^{-1} , which has been provided in the revised manuscript. In addition, the relationship between resistance and tensile strain of the electrode has provided in Supplementary Figure 1.

Comment #5: The authors should mention the output voltage and current of TENGs in the experiments (Figure 3 and Figure S5) because they can significantly vary with the applied pressure.

Response: Many thanks for your comments. The output voltage and current of the TENGs before and after 3 million cycles have been provided in the revised Supplementary Figure 6 (b) and (c), both of which demonstrate the stability of the TENG. In addition, the output voltage and current of TENGs under bending, twisting and lengthening motion have been provided in the revised Supplementary Figure 7.

Supplementary Figure 6. Stability of the TENG, (a) charge density as function of cycle number, inset shows the charge density of the TENG before and after 3 million cycles at 10 Hz, (b) open-circuit voltage and (c) short-circuit current before and after 3 million cycles at 10 Hz.

Supplementary Figure 7. Open-circuit voltage and short-circuit current of the TENG (a, b) under bending motion, (c, d) under twisting motion, (e, f) under lengthening motion.

Comment #6: In Figure 5, the authors showed charging of the LIBs by the "energy-shoe" while walking and jogging in the limited voltage range. However, the charging curve (Figure 5f) seems to be saturated at ~ 3.765 V. Therefore, an additional full charging curve of the LIBs in the voltage range of 2.75 - 4.24 V as shown in Figure S7 would be helpful.

Response: Thank you for your suggestion. The full charging curve of the LIBs has been provided in Supplementary Figure 11, where the LIB was charged by pressing 5 TENG-tubes connected in parallel at 10 Hz with a shaker.

Supplementary Figure 11. Charging-discharging curve of the LIB at a current density of 10 μA (Blue line) and charging by pressing 4-parallel TENG-tubes at 10 Hz then discharging at 10 μA (Red line).

Comment #7: The wearable energy harvesting and storage modules are typically connected with LEDs and/or sensors. Hopefully their monolithic integration with wearable LEDs/displays (e.g., Nature Communications 6, 7149, 2015; Science Advances 2, e1501856, 2016) and/or biosensors (e.g., Nature 529, 509, 2016; Nature Nanotechnology AOP, DOI: 10.1038/nnano.2016.38, 2016) will be a future research and development goal. These systems are still using rigid LIBs without self-charging and thereby frequently need to be replaced, although self-charging would dramatically enhance the portability. There have been reports that wearable LIBs or fabric-based SCs are charged by integrated TEGs to prolong the time of use. Some comments on this research direction in the introduction would be helpful to give guidance for the next research step.

Response: Thank you very much for your suggestion. Some related comments have added in the introduction, which are highlighted in the revised manuscript.

Reviewer #2 (Remarks to the Author):

The authors describe an interesting approach to fabricate wearable energy device through the triboelectric effects. They claimed that the tube-like design offers advantages including flexibility, stretchability, isotropy, weavability, and water-resistance. They claimed the highest surface charge density achieved. The biomechanics energy from walking or jogging appears to be sufficient to drive wearable electronics such as

electronic watch and fitness tracker. This work is an extension of the group's previous research. This review recommend publication after addressing the following issues:

Response: We appreciate the praise and the valuable comments of this reviewer.

(1) What is the durability of the devices? In particular, how will the long-term ambient moisture influence the device performance?

Response: Thank you for your comments. The durability of the TENG is tested by pressing the device 3 million cycles at 10 Hz. As shown by Supplementary Figure 6, the output performance of the TENG, including charging density, open-circuit voltage and short-circuit current remain very stable after 3 million cycles.

Generally, the ambient moisture will influence the TENG performance. Therefore, we design the water-resistance structure in the TENG, i.e. the two-ends of the TENG tube are enclosed by silicon rubber. As a result, the device can be immersed and washed in the water without the performance attenuation, as shown in Figure 4 (a) and Supplementary video 1. Furthermore, after immersion in water for 1 week, the output charge density still remains stable.

(2) The research group has reported their triboelectric devices in a number of previous publications. I wonder the key conceptual and methodological advances compared to their previous results.

Response: Thank you for your comments. In the present research, we are aiming to immediately and sustainably power wearable electronic devices with high-output triboelectric nanogenerator (TENG) and corresponding energy storage unit purely by harvesting biomechanical energy from daily motions. This is one big step of TENG towards practical use as an independent energy source and also the conceptual advances of the present work. In comparison, previous TENGs either failed to come up with sufficient electric output to sustainably powering electronic devices (Wang S., Lin L., et al. Nano letters, 2012, 12(12): 6339-6346; Xie Y., Wang S., et al. Advanced Materials, 2014, 26(38): 6599-6607; Chen J, Yang J, et al. ACS Nano, 2015, 9(3): 3324-3331.), or needed to be triggered by other machines, such as the radial-arrayed rotary TENG (Zhu G., Chen J., et al. Nature Communications, 2014, 5.) that harvested energy to power electronic devices from spinning motions of a high-speed spinning motor.

To achieve the goal discussed above, three principles need to be considered. (1) The TENG in this work should have ultra-high output performance; (2) It should be easy for the TENG to harvest biomechanical energy from daily human motions; (3) The self-powered system should be wearable. In dealing with those concerns, the methodological innovations have been summarized below. For high output performance, the authors

innovatively utilized super soft conductive and dielectric silicon rubber to be triboelectric layers. In this way, more area of the contact surface could be brought into intermolecular distance for triboelectrification (Li S., Zhou Y., et al. ACS Nano, 2016, 10(2): 2528-2535.). Moreover, the helix inner electrode of the tube separates the contact surface into small areas, which is also beneficial for the improvement of surface charge density (Supplementary Figure 3) and is for the first time presented. For harvesting biomechanical energy from daily motions of human being, all-elastomer-based flexible tube was designed and fabricated, it not only could extract mechanical energy from pressing, stretching, bending and twisting, but also could be weaved together. Furthermore, integrating the ultra-high-output TENG with supercapacitor or lithium-ion battery, wearable self-powered system has been built in the present research.

(3) what is the stretchability limit?

Response: Many thanks for your comments. The tensile stress-strain curve and the relationship between resistance and tensile strain have been provided in revised supplementary information. As shown in Supplementary Figure 1, the stretchability limit is 620%.

Supplementary Figure 1. Stretchability of the silicone rubber electrode. Tensile stress-strain curve (black curve) and Relationship between resistance and tensile strain (blue line+symbol).

(4) some typological errors, e.g. " $\square\text{Cm}^{-2}$ " in the abstract;

Response: Many thanks for your comments. The revised manuscript has been checked very carefully and some typological errors have been modified.

Reviewers' Comments:

Reviewer #1 (Remarks to the Author)

All comments from the reviewer were well addressed in the revised manuscript, which is now ready for publication.

Reviewer #2 (Remarks to the Author)

The authors have almost addressed all my concerns. As for the novelty argument, I would like to suggest that they should incorporate their statements into the revised manuscript to show the conceptual advances compared to their previous papers.

Respond to reviewers' comments:

Reviewer #1 (Remarks to the Author):

All comments from the reviewer were well addressed in the revised manuscript, which is now ready for publication.

Response: Thank you very much for your comments.

Reviewer #2 (Remarks to the Author):

The authors have almost addressed all my concerns. As for the novelty argument, I would like to suggest that they should incorporate their statements into the revised manuscript to show the conceptual advances compared to their previous papers.

Response: Thank you so much for your comments and suggestions. We have incorporated our statements in the revised manuscript.